# Age-Dependent Surface Receptor Expression Patterns in Immature Versus Mature Platelets in Mouse Models of Regenerative Thrombocytopenia

**DOI:** 10.3390/cells12192419

**Published:** 2023-10-08

**Authors:** Anita Pirabe, Sabine Frühwirth, Laura Brunnthaler, Hubert Hackl, Anna Schmuckenschlager, Waltraud C. Schrottmaier, Alice Assinger

**Affiliations:** 1Institute of Vascular Biology and Thrombosis Research, Center of Physiology and Pharmacology, Medical University of Vienna, 1090 Vienna, Austria; 2Institute of Bioinformatics, Biocenter, Medical University of Innsbruck, 6020 Innsbruck, Austria; hubert.hackl@i-med.ac.at

**Keywords:** aging, immature platelets, surface receptor expression, platelet function, age-related diseases

## Abstract

Aging is a multifaceted process that unfolds at both the individual and cellular levels, resulting in changes in platelet count and platelet reactivity. These alterations are influenced by shifts in platelet production, as well as by various environmental factors that affect circulating platelets. Aging also triggers functional changes in platelets, including a reduction in RNA content and protein production capacity. Older individuals and RNA-rich immature platelets often exhibit hyperactivity, contributing significantly to pathologic conditions such as cardiovascular diseases, sepsis, and thrombosis. However, the impact of aging on surface receptor expression of circulating platelets, particularly whether these effects vary between immature and mature platelets, remains largely unexplored. Thus, we investigated the expression of certain surface and activation receptors on platelets from young and old mice as well as on immature and mature platelets from mouse models of regenerative thrombocytopenia by flow cytometry. Our findings indicate that aged mice show an upregulated expression of the platelet endothelial cell adhesion molecule-1 (CD31), tetraspanin-29 (CD9), and Toll-like receptor 2 (TLR2) compared to their younger counterparts. Interestingly, when comparing immature and mature platelets in both young and old mice, no differences were observed in mature platelets. However, immature platelets from young mice displayed higher surface expression compared to immature platelets from old mice. Additionally, in mouse models of regenerative thrombocytopenia, the majority of receptors were upregulated in immature platelets. These results suggest that distinct surface receptor expressions are increased on platelets from old mice and immature platelets, which may partially explain their heightened activity and contribute to an increased thrombotic risk.

## 1. Introduction

Megakaryocytes are generated from hematopoietic stem cells in the bone marrow [1]. During maturation, they enlarge during endomitosis and accumulate RNA [2,3], ribosomes, and organelles [4,5]. Finally, proplatelet projections emerge, which release platelets into the circulation [6]. Newly formed platelets are rich in RNA and ribosomes, enabling them to synthesize proteins.

The lifespan of platelets in the bloodstream range from four to five days in mice and eight to nine days in humans [7,8]. Upon entering the circulation, platelets carry mRNA [2,9,10] during the first 24 h; hence, they are referred to as reticulated or immature platelets. These immature platelets constitute roughly 5% of the overall platelet population in humans [11] and about 7% in mice during steady conditions [12]. After the first day in circulation, the RNA content and protein synthesis capacity decline. Numerous studies have shown that immature platelets, which possess elevated levels of mRNA, exhibit increased reactivity [13,14]. This phenomenon is attributed to functional pathways and a higher protein content in immature platelets, features that are lost in older platelets [15]. Therefore, elevated counts of immature platelets are predictive markers for various pathologies, such as thrombocytopenia, infections, or sepsis [16,17,18].

Multiple studies have indicated that platelet activity increases also over an individual’s lifespan, suggesting a potential contribution to the development of vascular and thrombotic disorders in the elderly [19,20]. Platelet hyperactivity is defined by a reduced bleeding time, faster rate of clot formation [21,22], and increased sensitivity to agonist-induced aggregation [23,24,25]. The causes behind platelet hyperactivity in older individuals encompass elevated oxidative stress [26], alterations in plasma membrane structure [27], and changes in the platelet proteome [28] such as diminished prostaglandin I2 (PGI2) receptor quantities [29].

While most studies focus on platelet responsiveness, very little is known on the effect of aging on surface receptor expression at basal conditions. Therefore, we investigated the effect of aging on an individual and cellular level, by studying mature and immature platelets in young and aged mice as well as following platelet surface receptor expression over their lifespan in two different models of regenerative thrombocytopenia.

## 2. Material and Methods

### 2.1. Mouse Models and Platelet Depletion

All experiments and animal studies were conducted according to institutional guidelines and were approved by the Animal Care and Use Committee of the Medical University of Vienna (2022-0.792.111, 2023-0.184.589). Platelet depletion was implemented in mice with an inducible diphtheria toxin receptor (iDTR) under the regulation of the platelet-specific PF4 Cre recombinase (PF4 iDTR) as previously described [30]. This renders megakaryocytes susceptible to diphtheria toxin (DT), which induces megakaryocyte apoptosis in the bone marrow, resulting in thrombocytopenia. Platelet depletion in iDTR mice was achieved by subcutaneous injections of 100 ng DT (Merck Millipore) (Sigma-Aldrich, St. Louis, MI, USA) three times a week every second day (day -7, -5, and -3). In addition, we used a second mouse model in which platelet depletion was induced in wild type (WT) mice through antibody-mediated platelet removal from the bloodstream using intravenous administration of 1 μg/g anti-mouse glycoprotein (GP) Ibα (day -2) (R300, Emfret Analytics, Würzburg, Germany).

### 2.2. Blood Draw and Flow Cytometry Staining

Mice were anesthetized by isoflurane (Forane; Baxter Healthcare Corporation, Deerfield, IL, USA) and blood collected from the retro-orbital sinus was immediately anticoagulated with heparin (25 U/mL) and Acid–Citrate–Dextrose (ACD) (1:10). Overall hematological characteristics were evaluated using an automated hematology analyzer (Element HT5-Heska), while the number of immature (RNA-rich) platelets was determined through flow cytometry by staining platelets with anti-CD61 antibodies (20 min) and the Syto RNASelect dye (1 µM, 20 min). For platelet subset characterization, platelets were washed as previously described [31], stained with the respective antibodies (Appendix A) for 20 min, and fixed (1% formaldehyde). Samples were measured using a CytoFLEX flow cytometer and analyzed with the CytExpert 2.4 software (both from Beckman Coulter, Brea, CA, USA).

### 2.3. Immunohistochemistry

Bone marrow was isolated from murine femurs as previously described [30]. Bone sections (8 μm) were fixed in ice-cold acetone (5 min), permeabilized (0.5% Triton-X100), and stained with anti-CD41 (1:50) and nuclei were counterstained with Hoechst33342 (5 μg/mL; Invitrogen, Waltham, MA, USA) for 8 min (Appendix A). Tissues were incubated with a tissue autofluorescence quenching reagent for 5 min, mounted in antifade mounting medium (both Vector Laboratories, Newark, CA, USA), and imaged (Nikon A1plus Ti) (Nikon, Tokyo, Japan) at 20× magnification followed by analysis with ImageJ Fiji and CellProfiler 4.1.3.

### 2.4. Statistical Analysis

Statistical analyses were performed with GraphPad Prism 9; diagrams were generated with GraphPad Prism 9, Adobe Illustrator CS6 16.0.0, and BioRender. Hematological parameters and platelet receptor expression were compared using the Student’s T test after validation of normality with the Shapiro–Wilk test. If the data did not have a Gaussian distribution, groups were compared using the Mann–Whitney U test. Two-way ANOVA for multiple comparisons was performed for data with multiple predictors. The Šídák test was applied to test for multiple comparisons. *p*-values ≤ 0.05 were considered statistically significant. Different *p*-values *p* < 0.05, *p* < 0.01, *p* < 0.001, and *p* < 0.0001 are indicated as *, **, ***, and ****, respectively. Only two-sided tests were used. Principal component analyses were performed on standardized data using the R statistical software environment (v.4.3.0) package stats. First, two principal components were visualized and loadings for respective surface receptors were represented as arrows from the center.

## 3. Results

To study the impact of individual age on platelet turnover and platelet surface receptor expression, blood counts and platelets from young (10-weeks) and aged (50-weeks) WT mice were analyzed for surface expression of CD41, CD61, CD31, CD36, CD9, and activation markers CD40L, CD62P, CD63, and CD107a (Figure 1A). While hematological analysis revealed similar counts of red blood cells (RBCs) and white blood cells (WBCs) in young and old mice, the number of platelets (PLTs) and the mean platelet volume (MPV) were significantly elevated in old mice (Figure 1B). When we compared surface receptor expressions between young and old mice, we observed that the expression levels of CD31, CD9, and Toll-like receptor (TLR2) were significantly higher in old mice (Figure 1C), whereas CD36, CD40L, CD63, and CD107a were significantly reduced in old mice compared to young mice (Figure 1D). When comparing mature platelets to immature platelets, we observed that while surface receptor expressions did not differ between mature platelets from young and old mice, immature platelets from young mice exhibited significantly higher expression levels of some surface receptors including CD61, CD31, CD36, and the activation markers CD63 and CD107a (Figure 1E and Appendix A). These results suggest that the count and size of platelets increase with individual age but surface receptor expression is reduced in immature platelets from old mice, with mature platelets not showing any age-related differences in surface receptor expression patterns.

To further study surface expression in immature platelets, we expanded the immature platelet fraction by employing two different models of regenerative thrombocytopenia. Regenerative thrombocytopenia was either achieved by diphtheria toxin (DT)-induced apoptosis of megakaryocytes in the bone marrow of transgenic megakaryocyte specific iDTR mice (PF4 iDTR) or by anti-GPIbα antibody (R300)-mediated depletion of platelets in the circulation (Figure 2A). Following platelet depletion, blood samples were collected daily to monitor the recovery of platelet counts and platelet phenotype changes. Platelet counts began to rise 4 days after the final DT injection in iDTR mice and 3 days after R300 injection in WT mice, constituting day 1 of regenerative thrombocytopenia (data not shown), reaching around 30,000 and 170,000 PLTs/µL, respectively (Figure 2B). The MPV was significantly elevated in both mouse models, while iDTR also exhibited significantly higher WBCs but significantly lower RBCs after platelet depletion (Appendix A). We further monitored platelet counts and immature platelet fractions, starting before platelet depletion (day -7) until five days after platelet nadir (days 1–5), at which point, both mouse models had reached a peak in platelet count (Figure 2C). On day 1 of regenerative thrombocytopenia, the immature platelet fraction increased more than 8-fold from about 7% to over 55% because the total platelet count was more than 50-fold reduced due to the depletion of mature platelets and the majority of circulating platelets and therefore this fraction is represented by recently released platelets rich in RNA. From day 1 to day 5 post platelet nadir, a daily rise in absolute platelet counts was observed, which was associated with an overall decrease in the proportion of immature platelets. Of note, platelet counts in the R300 mouse models returned to physiological values, whereas an overproduction of platelets was observed in iDTR mice, which was associated with a second wave of immature platelets (Figure 2C). Next, we quantified megakaryocytes in the bone marrow of the iDTR and R300 mouse models as well as untreated mice and found that the number of megakaryocytes was elevated in both mouse models on the first day of regenerative thrombocytopenia compared to untreated mice (Figure 2D). These data demonstrate that mouse models of regenerative thrombocytopenia are suitable to expand the immature fraction of RNA-rich, immature platelets.

Next, we used mouse models of regenerative thrombocytopenia to study surface protein expression of immature (RNA-rich) and mature (RNA-poor) platelets on the day when the immature platelet fraction reached its peak (day 1). In PF4 iDTR mice, the vast majority of surface receptors (CD41, CD61, CD31, CD36, CD9, TLR2, and TLR9) (Figure 3A) and activation markers (CD62P, CD63, and CD107a) (Figure 3B) were significantly more abundant on immature than mature platelets. Only TLR4 and CD40L expression did not differ between the two platelet populations (Figure 3A,B). Also, the principal component analysis (PCA) of platelet subpopulations at all time points clustered separately in immature and mature platelet groups with only a small overlap (Figure 2C, left). While the mature platelet cluster was relatively homogenous, the immature platelets spread across Q1 and Q4. A more detailed PCA analysis displaying the day of regenerative thrombocytopenia indicates that the cluster of immature platelets on day 1 is clearly separated from immature platelets from day 2 to day 5. Mature platelets from day 1 also spread a bit across Q3, while mature platelets from day 2 to day 5 clustered nicely together (Figure 3D, right). Together these data imply that immature and mature platelets display different phenotypes as surface expression on immature platelets was higher, but also that immature and mature platelets from the first day of regenerative platelet production differ in their surface expression patterns from later stages of regeneration.

Finally, we also investigated platelet surface expression of immature and mature platelets after R300-mediated platelet depletion. On the first day of regenerative thrombocytopenia when the proportion of immature platelets was highest, the expression levels of almost all surface markers (CD41, CD61, CD31, CD9, TLR2, and TLR9) was significantly higher in immature platelets than mature platelets expect for TLR4 expression (Figure 4A), which mirrored the results obtained in the PF4 iDTR mice. Further, all the investigated activation markers (CD40L, CD62P, CD63, and CD107a) were significantly more abundant on the surface of immature compared to mature platelets (Figure 4B). PCA analysis of immature and mature platelets again reveals a clear separation of immature and mature platelet groups with just a small overlap (Figure 4C, left). While mature platelets spread a bit across Q1, immature platelets were separated in two smaller clusters (Figure 4C, left). When the days of regenerative thrombocytopenia were included in the PCA analysis, it was apparent that immature platelets on the first day are heterogeneous as they spread across Q2. Further, immature platelets from days 2 and 3 as well as immature platelets from days 4 and 5 clustered nicely together. In addition, mature platelets from day 1 also deviated from mature platelets from day 2 to day 5 (Figure 4C, right). To conclude, these data indicate that surface expression differs between immature and mature platelets and that immature platelets from day 1 are phenotypically distinguished from later stages according to higher surface receptor expression levels.

## 4. Discussion

In our study, we investigated the impact of aging on platelet surface receptor expression, examining both individual and cellular aspects. Our findings revealed that platelet counts and volumes increased as mice advanced in age. While mature platelets exhibited minimal alterations between their young and aged counterparts, a significant transformation in surface receptor expression levels was observed in immature platelets upon individual aging. To obtain further insights in the dynamics of platelet surface expression throughout the cellular lifespan, we used two regenerative platelet depletion models to increase the immature platelet fraction and allow for monitoring of these platelets as they age. In both models, we were able to identify a characteristic pattern of surface receptor expression in newly generated platelets. These findings provide valuable insights into the distinctive role of immature platelets and contribute to a deeper understanding of the underlying mechanisms.

Our findings on significantly higher platelet counts in aged mice compared to young mice are in line with the existing literature [32,33,34]. The age-associated increase in platelets could be a consequence of changes in the hematopoietic tissue in the bone marrow. Flach and colleagues found that hematopoietic stem cell numbers increase with age; however, they also detected a reduction in their functional activity and a general impairment of the hematopoietic system [35]. In addition, Poscablo and colleagues reported that the megakaryocyte cell progenitors displayed a higher proliferative potential and capacities to expand and reconstitute platelets [36,37]. This elevated platelet production could also contribute to the higher incidence of thrombotic events and cardiovascular disorders among the elderly [36].

Moreover, our investigation revealed that, although the majority of surface receptors were slightly reduced on platelets derived from aged mice, markedly elevated levels of the platelet endothelial cell adhesion molecule-1 (CD31 or PECAM-1), tetraspanin-29 (CD9), and TLR9 were detected on the surface of platelets from old compared to young mice. Elevated CD31, which facilitates platelet adhesion and aggregation at sites of endothelial injury [38], and increased levels of TLR9, which triggers platelet activation, degranulation, and aggregation during viral and bacterial infections [39,40], could contribute to dysregulated platelet functions in older individuals. On the contrary, upregulation of CD9, which is physically linked to GPIIb/IIIa (CD41/CD61) and prevents excessive thrombus growth, could represent a counterbalancing antithrombotic mechanism in aged mice [41]. Further, we found significantly lower expression levels of activation markers including CD40L, CD63, and CD107a in old compared to young mice, while a human study observed a higher basal activation in the elderly. However, enhanced platelet activation in older people has been associated with chronic inflammation [42], whereas our mice were not in an inflamed state, indicating that aging itself might not lead to increased platelet activation. Overall, murine and human studies revealed that aging is associated with increased platelet reactivity, reduced bleeding time, and higher susceptibility to venous and carotid artery thrombosis [33,43,44]. In addition, age-related changes in the mitochondrial mass and activity can modulate platelet functions. Platelets depend on mitochondria as they require adenosine triphosphate (ATP) to carry out their functions, such as adhesion, granule secretion, and aggregation [45,46]. Aging is associated with a decline in mitophagy, which impairs the removal of mitochondria, and elevated levels of pro-inflammatory and pro-thrombotic molecules, such as tumor necrosis factor α (TNF-α). All these factors, including age-related diseases such as diabetes, have been associated with an increase in the platelet mitochondrial mass, enhanced respiration, as well as platelet hyperactivity [47,48,49] and are associated with cardiovascular and neurodegenerative diseases [50,51,52]. Of note, mitochondria are involved in many processes beyond energy production, such as the generation of reactive oxygen species (ROS) and regulation of Ca+ signaling, which boost platelet activation or apoptosis, leading to phosphatidylserine (PS) expression on the platelet surface, which acts as a catalytic site for enzymes and thrombin generation, provoking faster platelet aggregation [53,54,55]. Moreover, mitochondria are physically linked to other organelles such as the plasma membrane and rough endoplasmic reticulum (ER), which modulate the synthesis and transport of lipids [56,57,58]. Thus, mitochondrial dysregulations could also result in changes in surface receptor expression pattern. Given the complexity of age-related changes in platelet physiology, several factors are likely to function together to contribute to platelet hyperactivity in the elderly.

Further, studies of megakaryocyte progenitors and platelets revealed different gene expression programs and transcriptomes in the elderly that are implicated in cell–cell signaling, platelet activation, and inflammatory pathways [36,59,60]. However, there is a need for further investigations to elucidate how thrombopoiesis and cellular aging of platelets affects the surface receptor expression pattern in young and old individuals. Our data add to the understanding of age-related functional changes of platelets, as we can show for the first time that the immature fraction specifically exhibits changes in their surface receptor expression patterns. These findings strongly suggest that immature platelets may play a central role in age-related thrombotic events, shedding light on a critical aspect of platelet biology in aging.

To gain a comprehensive understanding of the transformations that transpire throughout a platelet’s lifespan, we employed two distinct synchronization methods. In one approach, we depleted peripheral platelets, while in the other, we triggered apoptosis of megakaryocytes. In both scenarios, we observed a marked decline in platelet counts followed by a robust upsurge in the immature platelet fraction.

Similar to platelets in aged mice, we also detected elevated MPVs in our mouse models of regenerative thrombocytopenia. This is probably a consequence of aggravated thrombopoiesis, which is also observed in thrombocytopenic conditions [61] and was previously described in the iDTR mouse model [12]. In addition, PF4 Cre iDTR mice exhibited anemia; however, red blood cells are not expected to be directly affected by DT treatments, as they do not express the DT receptor. We therefore hypothesize that the DT-mediated depletion of megakaryocytes changes the bone marrow environment/niche and/or impairs cell–cell communication, which may affect erythropoiesis [62,63]. As the PF4 Cre model was previously described to also regulate gene expression in subsets of erythrocytes and white blood cells [64], we cannot rule out contributions of DT-mediated apoptosis of erythroid progenitor cells. However, given the murine erythrocyte lifespan of about 41 days and the fraction of RBCs expressing PF4 Cre [65], the observed decline in RBC counts far exceeds levels that could be explained by leaky Cre expression. Moreover, iDTR mice exhibited increased levels of white blood cells following depletion, probably due to the initial production of anti-DT antibodies [30].

Investigations of immature and mature platelets in mouse models of regenerative thrombocytopenia suggest that cellular aging affects the surface expression of certain receptors including CD41, CD61, CD31, CD36, CD9, TLR2, TLR9, CD62P, CD63, and CD107a (and CD40L in the R300 mouse model) on platelets. So far, there is only little information about receptor expression in immature and mature platelets in circulation, as most studies focus on platelet reactivity in vitro. The existing literature investigating the expression of surface markers suggests that immature platelets from mice express more CD31, which is in line with our findings. However, the same study also reported diminished CD9 expression and observed no alterations in GPIIb (CD41) expression [66], while we and others detected elevated levels of GPIIb (CD41) and GPIIIa (CD61) in the immature platelet fraction [67].

Consistent with the notion that immature platelets are characterized by heightened reactivity, our study revealed upregulation of all tested activation markers in immature platelets in both regenerative thrombocytopenia models. It is worth noting that previous reports have presented conflicting findings, with one study suggesting a 35% reduction in P-selectin (CD62P) expression on immature platelets compared to mature platelets [68]. Conversely, other studies did not observe any differences in the expression of P-selectin on immature and mature human [67] or murine platelets [66], further emphasizing the complexity and variability in platelet activation marker expression across different studies and contexts. In a broader context, our study highlights that surface receptor expression on platelets decreases as they age. This aligns with the discovery by Bernlochner and colleagues, who reported a 45% reduction in total protein content in older platelets compared to their younger counterparts [15]. These findings may signify the heightened functionality of aged platelets, rendering them more susceptible to activation and their active involvement in hemostasis and thrombosis [69,70,71]. Future proteomic and transcriptomic analysis of mature and immature platelets are warranted to shed light on the underlying changes in platelet populations during aging.

## 5. Conclusions

In summary, our study represents the first comprehensive exploration of platelet surface receptor expression at both the individual and cellular levels under resting conditions. Our findings suggest that, while platelets from older mice exhibit elevated expression of certain receptors that may promote hyperactivity, immature but not mature platelets from aged mice display a reduced surface expression pattern compared to their younger counterparts. Moreover, our investigations using mouse models of regenerative thrombocytopenia, which are conducive to expanding the immature platelet fraction, indicate a broad spectrum of increased receptor expression levels on immature platelets. While this initial study provides valuable insights into characterizing platelet surface expression in resting platelets, further investigations are warranted to delve into potential associations between elevated receptor surface expression pattern and platelet reactivity.

## Figures and Tables

**Figure 1 cells-12-02419-f001:**
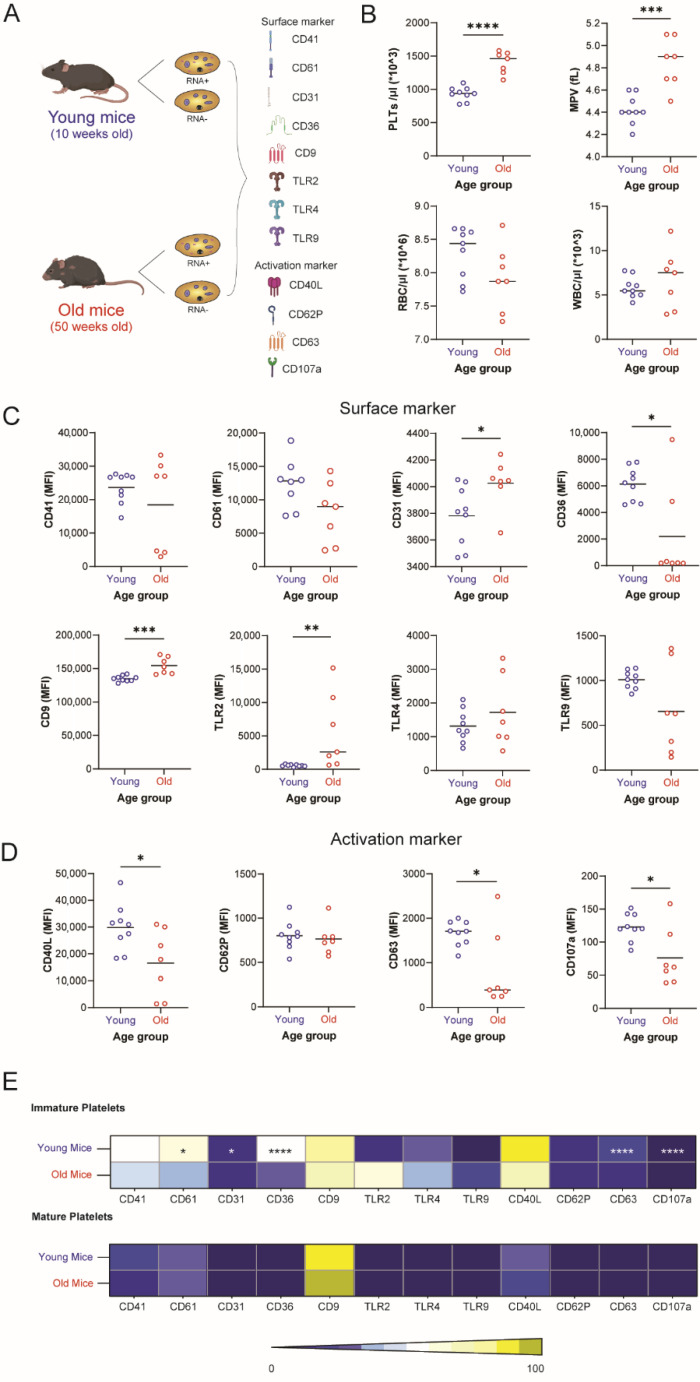
Comparison of platelet receptor expression according to the individual and cellular age. (**A**) Experimental setup: Blood was drawn from young (10-week) and old (50-week) mice for investigation of surface receptors (CD41, CD61, CD31, CD36, CD9, TLR2, TLR4, TLR9) and activation markers (CD40L, CD62P, CD63, CD107a) between young and old individuals as well as immature (RNA-rich) and mature (RNA-poor) platelets. (**B**) The number of platelets (PLTs), red blood cells (RBCs), white blood cells (WBCs), and the mean platelet volume (MPV) were determined in naïve mice. (**C**,**D**) PRP was stained for different platelet surface receptors (**C**) and activation markers (**D**) and analyzed by flow cytometry. Figures display receptor abundance on the surface of platelets from young and old mice. (**E**) PRP was stained with Syto RNASelect to identify immature (RNA-rich) platelets and their expression of surface and activation markers was measured by flow cytometry. Heatmap illustrates surface receptor expression of immature (upper) and mature (lower) platelets of young and old mice. The measure of central tendency denotes the mean (**B**–**D**). Different *p*-values *p* < 0.05, *p* < 0.01, *p* < 0.001, and *p* < 0.0001 are indicated as *, **, ***, and ****, respectively.

**Figure 2 cells-12-02419-f002:**
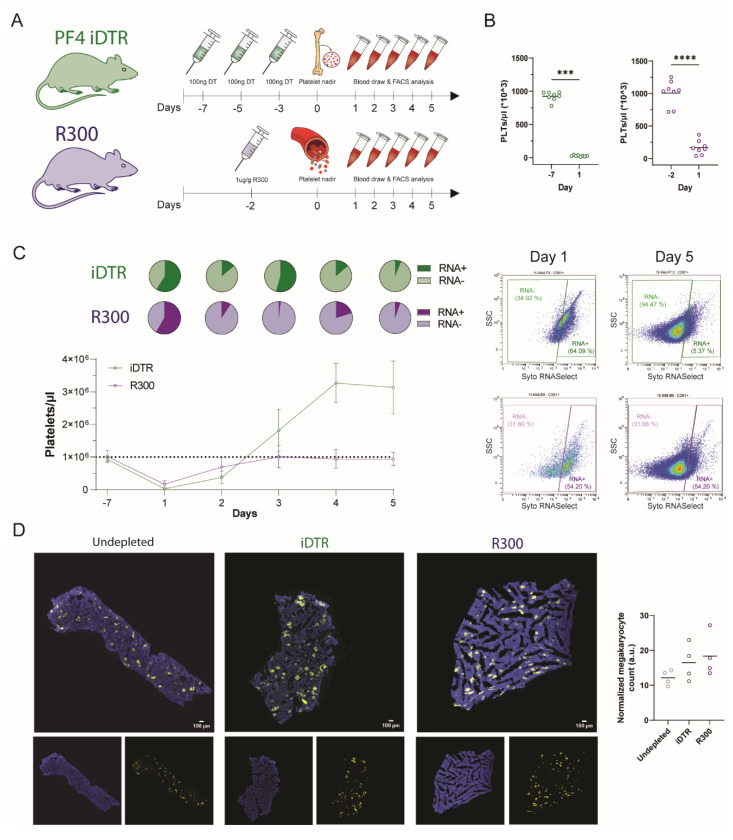
Mouse models of regenerative thrombocytopenia for the investigation of receptor expression on immature and mature platelets. (**A**) Experimental setup: Platelets were either depleted by diphtheria toxin (DT)-induced apoptosis of megakaryocytes in the bone marrow of PF4 iDTR mice or by antibody-mediated depletion (R300) of platelets in the bloodstream. After platelet nadir, blood was drawn on five consecutive days to investigate the platelet phenotype of immature and mature platelets. (**B**,**C**) Blood was drawn and analyzed with an automated hematology analyzer to determine the platelet counts (**B**,**C**) and whole blood was stained with anti-CD61 antibodies as well as Syto RNASelect to assess the number of immature platelets (**C**). The line chart displays the course of total platelet counts before (−7) and five days after platelet nadir (1–5) and pie charts show the percentage of immature (RNA-rich) and mature (RNA-poor) platelets (**left**). Flow cytometry plots show representative pictures of the RNA-positive population of immature platelets one and five days after platelet nadir (**right**) (**C**). (**D**) Femurs were harvested and bone sections were stained for CD41 (yellow) and Hoechst (blue) to quantify megakaryocytes after DT— or R300-mediated platelet depletion and untreated mice. Representative immunohistochemistry pictures are given. (**D**) Scatter diagrams represent the number of megakaryocytes of untreated mice and after platelet depletion normalized to the tissue size. The measure of central tendency denotes the mean (**B**–**D**) and error bars the SD (**B**). Different *p*-values *p* < 0.001, and *p* < 0.0001 are indicated as ***, and ****, respectively.

**Figure 3 cells-12-02419-f003:**
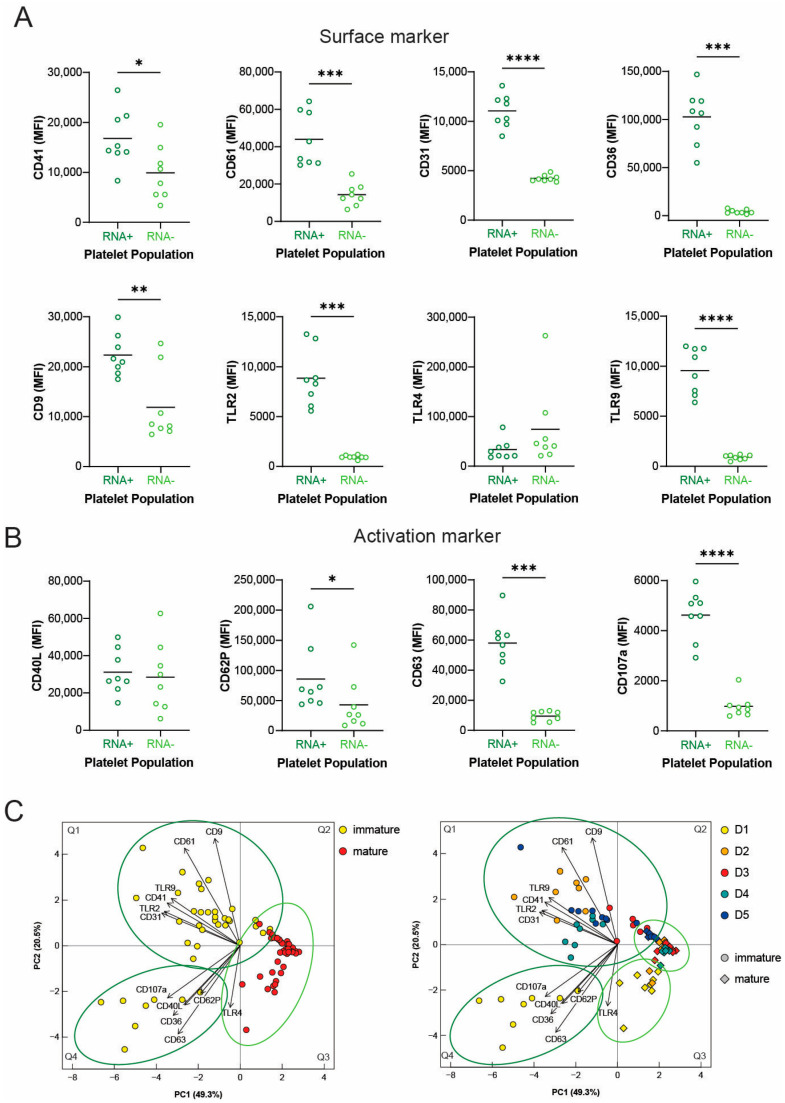
Surface receptor and activation marker expression on immature and mature platelets after DT-mediated platelet depletion. (**A**,**B**) PRP was stained with Syto RNASelect to identify immature platelets and distinct surface (**A**) and activation markers (**B**) were analyzed by flow cytometry. Scatter diagrams present the abundance of selected receptors on the surface of immature (RNA-rich) (dark green) and mature (RNA-poor) (light green) platelets on the first day of regenerative thrombocytopenia. (**C**) Principal component analyses (PCAs) of expression levels of immature and mature platelets at various time points. PCA plots visualize the clustering pattern of immature (yellow) and mature (red) platelets from day 1 to day 5 altogether (**left**) or separately according to the day post platelet nadir (day 1 = yellow; day 2 = orange; day 3 = red; day 4 = cyan; D5 = blue) (**right**). The measure of central tendency denotes the mean (**A**,**B**). Different *p*-values *p* < 0.05, *p* < 0.01, *p* < 0.001, and *p* < 0.0001 are indicated as *, **, ***, and ****, respectively.

**Figure 4 cells-12-02419-f004:**
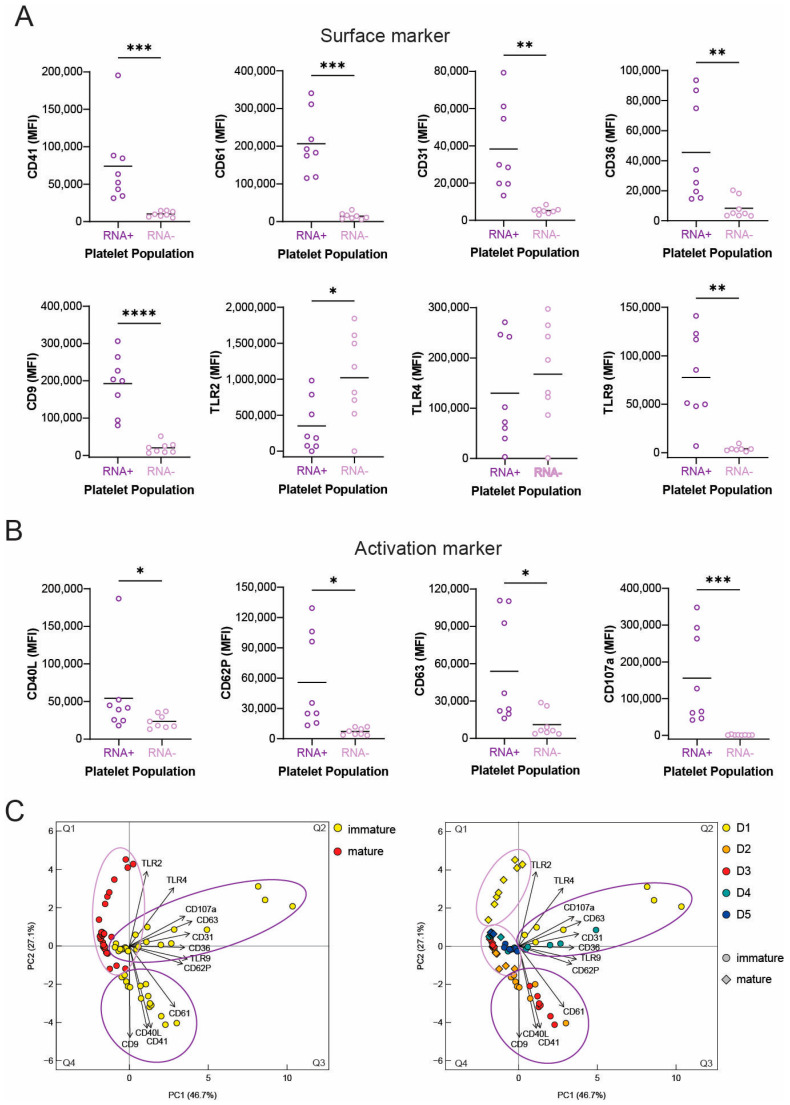
Surface receptor and activation marker expression on immature and mature platelets after R300-mediated platelet depletion. (**A**,**B**) PRP was stained with Syto RNASelect to identify immature platelets and distinct surface (**A**) and activation markers (**B**) were analyzed by flow cytometry. Scatter diagrams present the amount of selected markers expressed on the surface of immature (RNA-rich) (dark purple) and mature (RNA-poor) (light purple) platelets on the first day regenerative thrombocytopenia. (**C**) Principal component analyses (PCAs) of expression levels of immature and mature platelets at various time points. PCA plots visualize the clustering pattern of immature (yellow) and mature (red) platelets from day 1 to day 5 altogether (**left**) or separately according to the day post platelet nadir (day 1 = yellow; day 2 = orange; day 3 = red; day 4 = cyan; D5 = blue) (**right**). The measure of central tendency denotes the mean (**A**,**B**). Different *p*-values *p* < 0.05, *p* < 0.01, *p* < 0.001, and *p* < 0.0001 are indicated as *, **, ***, and ****, respectively.

## Data Availability

The data that support the findings of this study are available from the corresponding author upon reasonable request.

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
