# Peer review of "Age-Dependent Surface Receptor Expression Patterns in Immature Versus Mature Platelets in Mouse Models of Regenerative Thrombocytopenia"

_cells, 2023, doi:10.3390/cells12192419_

Round 1
Reviewer 1 Report
This is a nicely done survey study investigating surface receptor expression in immature and mature platelets, in young and old WT mice. The association of aging and increased risk of platelet-driven thrombosis is well known, and this study adds new information on molecular changes in aging platelets that add to our understanding of the functional and morphological changes in platelets as individuals age.
There are just a few minor concerns.
-
The authors separate platelets in “RNA+” and “RNA-” cohorts. It would be more accurate to describe these populations as RNA-rich and RNA-poor. It is unlikely that the “RNA-” platelets are truly depleted of RNA, including not just mRNA but other types of RNA that are enriched in platelets.
-
Recent work from several groups has identified increased mitochondrial mass and oxygen consumption in aging platelets, as a primary mechanism of platelet hyperreactivity in aged individuals. The authors are suggested to place their results in the context of these prior findings.
-
The increased platelet counts and increased mean platelet volumes in older mice raise the question of what is responsible for this large increase in platelet biomass, and also how this observation can be reconciled with the altered surface receptor expression. As noted above, increased mitochondrial mass may partially account for these differences. Given the observed reduction in surface expression of certain platelet receptors, is this simply an effect of dilution of a similar level of protein expression in megakaryocytes, spread out over a large number of platelets of increased size? A straightforward analysis of bone marrow megakaryocytes with flow cytometry under the different conditions would provide mechanistic insight to explain the differences in platelet receptor expression.
Author Response
This is a nicely done survey study investigating surface receptor expression in immature and mature platelets, in young and old WT mice. The association of aging and increased risk of platelet-driven thrombosis is well known, and this study adds new information on molecular changes in aging platelets that add to our understanding of the functional and morphological changes in platelets as individuals age.
There are just a few minor concerns.
- The authors separate platelets in “RNA+” and “RNA-” cohorts. It would be more accurate to describe these populations as RNA-rich and RNA-poor. It is unlikely that the “RNA-” platelets are truly depleted of RNA, including not just mRNA but other types of RNA that are enriched in platelets.
Reply: We agree with the reviewer and have changed RNA+ to RNA-rich and RNA- to RNA-poor.
- Recent work from several groups has identified increased mitochondrial mass and oxygen consumption in aging platelets, as a primary mechanism of platelet hyperreactivity in aged individuals. The authors are suggested to place their results in the context of these prior findings.
Reply: We thank the reviewer for his/her suggestion and we added this aspect in the discussion.
- The increased platelet counts and increased mean platelet volumes in older mice raise the question of what is responsible for this large increase in platelet biomass, and also how this observation can be reconciled with the altered surface receptor expression. As noted above, increased mitochondrial mass may partially account for these differences. Given the observed reduction in surface expression of certain platelet receptors, is this simply an effect of dilution of a similar level of protein expression in megakaryocytes, spread out over a large number of platelets of increased size? A straightforward analysis of bone marrow megakaryocytes with flow cytometry under the different conditions would provide mechanistic insight to explain the differences in platelet receptor expression.
Reply: Indeed, this is a very interesting question. Unfortunately, highly polyploidy megakaryocytes always ruptured during the isolation process and we were never retrieve mature megakaryocytes. So we are unfortunately currently unable to address this and we are not aware that anyone managed to look at this. However, we added a paragraph to the discussion.
Reviewer 2 Report
In this manuscript Pirabe et al investigated changes of certain surface expression in platelets during aging. They also included two different mouse models of regenerative thrombocytopenia for analysis of surface receptor expression during platelet maturation. It is well-known that platelet activity increases during aging, however most studies on this field focus on platelet responsiveness and comprehensive examination of surface receptor expression changes in individual platelets and in platelet population during aging were mostly not known. In this manuscript shown increased platelet volumes and counts in mice during aging and that most significant changes occurs in immature platelets population. This manuscript could be the basis for future proteomic and transcriptomic analysis of mature and immature platelets and whole platelet population during aging. The manuscript is very well written, all experiments and presented data are clear and highly significant for understanding the mechanisms of increased platelet reactivity in older individuals.
In the whole manuscript, I found only one place that should be corrected (p. 5, l 169 From 1 one to day 5 post).
Author Response
In this manuscript Pirabe et al investigated changes of certain surface expression in platelets during aging. They also included two different mouse models of regenerative thrombocytopenia for analysis of surface receptor expression during platelet maturation. It is well-known that platelet activity increases during aging, however most studies on this field focus on platelet responsiveness and comprehensive examination of surface receptor expression changes in individual platelets and in platelet population during aging were mostly not known. In this manuscript shown increased platelet volumes and counts in mice during aging and that most significant changes occurs in immature platelets population. This manuscript could be the basis for future proteomic and transcriptomic analysis of mature and immature platelets and whole platelet population during aging. The manuscript is very well written, all experiments and presented data are clear and highly significant for understanding the mechanisms of increased platelet reactivity in older individuals.
Reply: We thank the reviewer for his/her positive response and added an outlook paragraph regarding future proteomic and transcriptomic analysis of mature and immature platelets during aging in the discussion section.
In the whole manuscript, I found only one place that should be corrected (p. 5, l 169 From 1 one to day 5 post).
Reply: We thank the reviewer for pointing out this mistake, which has been corrected in the revised version.
Round 2
Reviewer 1 Report
The authors responded to the prior critiques.